# Isotropic 3D topological phases with broken time reversal symmetry

Hélène Spring[1], Anton R. Akhmerov[1], Dániel Varjas[2,3,4,5*]

**1** Kavli Institute of Nanoscience, Delft University of Technology, P.O. Box 4056, 2600 GA Delft, The Netherlands
**2** Department of Physics, Stockholm University, AlbaNova University Center, 106 91 Stockholm, Sweden
**3** Max Planck Institute for the Physics of Complex Systems, Nöthnitzer Strasse 38, 01187 Dresden, Germany
**4** IFW Dresden and Würzburg-Dresden Cluster of Excellence ct.qmat, Helmholtzstr. 20, 01069 Dresden, Germany
**5** Department of Theoretical Physics, Institute of Physics, Budapest University of Technology and Economics, Műegyetem rkp. 3., 1111 Budapest, Hungary
*varjas.daniel@ttk.bme.hu

## Abstract

**Axial vectors, such as current or magnetization, are commonly used order parameters in time-reversal symmetry breaking systems. These vectors also break isotropy in three dimensional systems, lowering the spatial symmetry. We demonstrate that it is possible to construct a three-dimensional medium with average isotropy and inversion symmetry where time-reversal symmetry is systematically broken. We devise a model of an amorphous material with scalar time-reversal symmetry breaking, implemented by hopping through chiral magnetic clusters along the bonds. The presence of only average spatial symmetries—continuous rotation and inversion—is sufficient to protect a topological phase, yielding a statistical topological insulator. We demonstrate the topological nature of our model by constructing a bulk integer topological invariant for the effective continuum model, which guarantees gapless surface spectrum on any surface with an odd number of Dirac nodes, analogous to crystalline mirror Chern insulators. We also show the expected transport properties of a three-dimensional statistical topological insulator, which remains critical on the surface for odd values of the invariant.**

## 1 Introduction

A three-dimensional (3D) isotropic medium has the highest degree of spatial symmetry, invariant under all rotations and inversion. Unless they are explicitly broken, non-spatial symmetries like time-reversal symmetry (TRS) are also present in isotropic systems. Removing TRS typically also breaks isotropy, for example ferromagnets break TRS but also break rotation symmetry along the axes which are not parallel to the magnetization. Antiferromagnets restore some spatial symmetries such as the product of inversion and TRS, but also break rotation symmetry [1]. The spatial symmetries are partially restored in altermagnets [2]—a recently proposed class of materials combining lack of net magnetization with a spin splitting away from away from high-symmetry momenta, however even

in these materials the magnetic order is incompatible with full isotropy. Related questions also have been studied in the context of symmetry classification of non-collinear antiferromagnetic orders, identifying toroidal magnetic monopoles as time-reversal breaking configurations compatible with a high level of magnetic space group symmetry [3, 4], however, the question of spatial isotropy was not addressed.

The spatial symmetries of a system are relevant both for defining and protecting topological phases [5–8]. While initially considered to be susceptible to disorder, topological systems relying on spatial symmetries were later shown to be protected from localization as long as the disordered ensemble respects the spatial symmetries [9–11]. Interacting symmetry-protected topological phases protected by a combination of average and exact symmetries have also been found in recent studies [12, 13]. The protection by an average symmetry, a hallmark of statistical topological insulators, is especially powerful in amorphous media that naturally possess isotropy on average. In an earlier work we demonstrated that unlike their crystalline counterparts—where the spatial symmetry is only preserved by certain crystal terminations—it is possible to utilize the isotropy of a 2D amorphous medium to extend the topological protection to any edge of the system [14].

Motivated by the two above considerations, we ask whether it is possible to find a model hosting a non-interacting topological phase protected only by average continuous spatial symmetries. Because both TRS and average TRS protect topological phases, we additionally require that the desired model also breaks TRS on average. By designing a scalar, rather than a vector TRS breaking order using a random assembly of chiral magnetic molecules, we answer positively to the above question. Specifically we demonstrate that the average spatial symmetries present in 3D isotropic media protect topological phases even when TRS is systematically broken, and that the amorphous realization of such a system is a statistical topological insulator. This topological phase is analogous to crystalline mirror-Chern insulators, except that the isotropic system hosts gapless modes on any flat surface regardless of orientation. Furthermore, we identify a bulk higher-order electromagnetic response which distinguishes isotropic media with or without scalar TRS breaking.

The organization of the manuscript is as follows. In Sec. 2 we formulate an isotropic continuum model where TRS is systematically broken. We present a microscopic Hamiltonian originating from an amorphous network of chiral magnetic molecules that replicates this model. In Sec. 3 we demonstrate the topological nature of our model by formulating bulk invariants, examining surface dispersions, and analyzing transport of the topologically protected surface modes. As established in the study of statistical topological insulator phases, we show that the model localizes when its degrees of freedom are doubled. We conclude in Sec. 4.

## 2   Symmetry analysis

### 2.1   Continuum model

In order to guide the construction of a microscopic model, we begin by developing a minimal continuum ($\boldsymbol{k} \cdot \boldsymbol{p}$) model respecting the desired symmetries. We use the method of invariants [15], a systematic approach to construct $\boldsymbol{k} \cdot \boldsymbol{p}$ Hamiltonians respecting a set of symmetry constraints. While this method can be carried out by hand, it becomes involved for a large number of bands and high orders in $\boldsymbol{k}$, so we automate this process using the software package Qsymm [16]. A generic spatial symmetry group element $g$ imposes a

constraint on the continuum Hamiltonian $H(\boldsymbol{k})$ of the form

$$U_g H(\boldsymbol{k}) U_g^\dagger = H(R_g \boldsymbol{k}), \tag{1}$$

where $R_g$ is the orthogonal spatial transformation matrix of real and $\boldsymbol{k}$-space, and $U_g$ is its representation on the internal Hilbert space. The specific form of $U_g$ depends on the underlying degrees of freedom, (e.g. the spin and orbital character of the bands included in the $\boldsymbol{k} \cdot \boldsymbol{p}$ Hamiltonian), the only restriction being that they form a consistent (double) representation of the symmetry group.

We specifically want to examine systems invariant under all continuous rotations and inversion, a group isomorphic to $O(3)$, which also includes all mirrors as combinations of a twofold rotation and inversion. A generic pure rotation is characterized by a rotation vector $\boldsymbol{n}$, and its real-space and unitary action can be written as

$$R_{\boldsymbol{n}} = \exp(-i\boldsymbol{n} \cdot \boldsymbol{L}), \qquad U_{\boldsymbol{n}} = \exp(-i\boldsymbol{n} \cdot \boldsymbol{S}), \tag{2}$$

where $\boldsymbol{S}$ is a vector of internal angular momentum operators, and $\boldsymbol{L}$ the a vector of 3D spatial rotation generators, both obeying angular momentum commutation relations. Substituting these into the symmetry constraint, and taking the derivative with respect to $n_i$ yields

$$[H(\boldsymbol{k}), S_i] = \left(\frac{\partial H}{\partial \boldsymbol{k}}\right)^T L_i \boldsymbol{k}. \tag{3}$$

Inversion symmetry imposes the constraint

$$U_\mathcal{I} H(\boldsymbol{k}) U_\mathcal{I}^\dagger = H(-\boldsymbol{k}), \tag{4}$$

where $U_\mathcal{I}^2 = \mathbb{1}$ and $[U_\mathcal{I}, S_i] = 0$ in order to form a consistent representation. For a given symmetry representation characterized by a set of $\boldsymbol{S}$ and $U_\mathcal{I}$, we search for $H(\boldsymbol{k})$ in the form of a power series with unknown matrix coefficients, and solve the above equations order-by-order to find the most generic parametric family of symmetry-allowed Hamiltonians. The inverse problem can also be solved: given a family of Hamiltonians, it is possible to find a complete set of symmetry generators of the above form, as well as time-reversal and particle-hole type symmetries. [16]

We follow the procedure outlined in Ref. [14]: We systematically construct inequivalent nontrivial representations of the symmetry group with increasing matrix dimensions, generate 3D $\boldsymbol{k}$-linear (Dirac) Hamiltonians, and check whether the bulk is gappable, i.e. whether a $\boldsymbol{k}$-independent constant term is allowed. We find that the smallest such Hamiltonian has 4-bands with the form

$$H_{\mathrm{Dirac}}(\boldsymbol{k}) = \mu_0 \sigma_0 \tau_0 + \mu' \sigma_0 \tau_z - t_0 \boldsymbol{\sigma} \cdot \boldsymbol{k} \tau_x - t_1 \boldsymbol{\sigma} \cdot \boldsymbol{k} \tau_y, \tag{5}$$

with symmetry representations

$$U_\mathcal{I} = \sigma_0 \tau_z, \ \ S_x = \frac{1}{2} \sigma_x \tau_0, \ \ S_y = \frac{1}{2} \sigma_y \tau_0, \ \ S_z = \frac{1}{2} \sigma_z \tau_0, \tag{6}$$

where $\boldsymbol{\sigma}$ and $\boldsymbol{\tau}$ are two sets of Pauli matrices, corresponding to spin and orbital degrees of freedom. The two types of orbitals both transform under a spin-1/2 irreducible representation, but have different parity under inversion. Bands with such symmetry characters can arise for example from a spinful $s$-orbital, and the $J = 1/2$ subspace of a spin-orbit split $p$-orbital.

In the next step we make sure that the model has no symmetries beyond the spatial isotropy, because additional symmetries would make it impossible to assign topological

protection to the spatial symmetries only. We find that the Dirac Hamiltonian above does have an additional time-reversal symmetry $\mathcal{T} = \sigma_y \exp(-i\tau_z \phi)\mathcal{K}$ where $\tan\phi = t_1/t_0$ and $\mathcal{K}$ is complex conjugation. In order to break this symmetry, we allow higher-order terms, including up to cubic momentum dependence, resulting in a continuum model of the form

$$
\begin{aligned}
H_{4\times4}(\boldsymbol{k}) = {} & (\mu_1 + t_2 k^2)\sigma_0(\tau_0 + \tau_z)/2 + (\mu_2 + t_3 k^2)\sigma_0(\tau_0 - \tau_z)/2 \\
& + (-t_1 + t_4 k^2)\boldsymbol{\sigma} \cdot \boldsymbol{k}\tau_y + (-t_0 + t_5 k^2)\boldsymbol{\sigma} \cdot \boldsymbol{k}\tau_x,
\end{aligned}
\tag{7}
$$

where we interpret $\mu_i$ as chemical potentials, and $t_i$ as normal and spin-orbit hopping amplitudes. We demand that $t_4/t_1 \neq t_5/t_0$, otherwise a momentum-dependent time-reversal symmetry of the form $\mathcal{T} = \sigma_y \exp(-i\tau_z \phi(\boldsymbol{k}))\mathcal{K}$ would still exist. This ensures that the phase of the spin-orbit terms connecting the two types of orbitals is momentum-dependent, such terms originate from distance-dependent hopping phases in the tight-binding models.

Despite lacking TRS, the high degree of spatial symmetry of this model protects the twofold spin degeneracy of all bands. For a fixed $\boldsymbol{k}$, the eigenstates of (7) are eigenstates of the angular momentum operator $\boldsymbol{k} \cdot \boldsymbol{S}$ in the direction parallel to $\boldsymbol{k}$. Mirror symmetries that leave $\boldsymbol{k}$ invariant exchange states with opposite angular momentum, thereby ensuring the degeneracy of the spin bands.

Finally we check that the symmetry is capable of protecting gapless surface states. We restrict the symmetry group to the subgroup that leaves a flat surface invariant. In the case of a surface with normal $\hat{z}$, this group is generated by rotations around $\hat{z}$ and a mirror with normal perpendicular to $\hat{z}$, for example $\mathcal{M}_x$. Considering a 2-band model in 2D with symmetry representations for rotation by angle $\phi$ as $U_\phi = \exp(-i\phi\sigma_z)$ and for $\mathcal{M}_x$ as $U_{\mathcal{M}_x} = i\sigma_x$, we find the symmetry-allowed massless Dirac Hamiltonian

$$
H_{2D}(k_x, k_y) = \mu\sigma_0 + t(k_x\sigma_y + k_y\sigma_x).
\tag{8}
$$

The combination of continuous rotation and mirror symmetries is sufficient to forbid a $k$-independent mass term from opening a gap in single surface Dirac cone. We emphasize the role of the mirror symmetries originating from the bulk inversion symmetry, rotation invariance alone would allow for a mass term proportional to $\sigma_z$ to open a gap.

## 2.2 Amorphous realization

Amorphous systems typically possess average continuous rotation symmetry, average reflection and average inversion, unless symmetry-breaking external fields are present. This symmetry is not only manifest in the structure, but also in the Hamiltonian as we argue below. We construct short-range correlated amorphous structures using the same procedure as in Ref. [14], treating sites as hard spheres, and connecting nearby sites with hoppings, resulting in a graph embedded in 3D space to host the tight-binding Hamiltonian. We treat this amorphous structure as a quenched background disorder, and do not concern ourselves with its origin.

In gapped solids the effective tight-binding Hamiltonian is in general a local and symmetric function of the disorder configuration [8, 17]. For simplicity, we further assume that there is only one type of atom in the system, the onsite terms are constant, while the hopping terms only depend on the hopping vector $\boldsymbol{d}$, allowing to write the amorphous tight-binding Hamiltonian in the form

$$
H = \sum_{\boldsymbol{r},i,j} H_{ij}^{\text{onsite}} c_{\boldsymbol{r},i}^\dagger c_{\boldsymbol{r},j} + \sum_{\langle \boldsymbol{r},\boldsymbol{r}'\rangle,i,j} H_{ij}^{\text{hop}}(\boldsymbol{r} - \boldsymbol{r}') c_{\boldsymbol{r},i}^\dagger c_{\boldsymbol{r}',j},
\tag{9}
$$

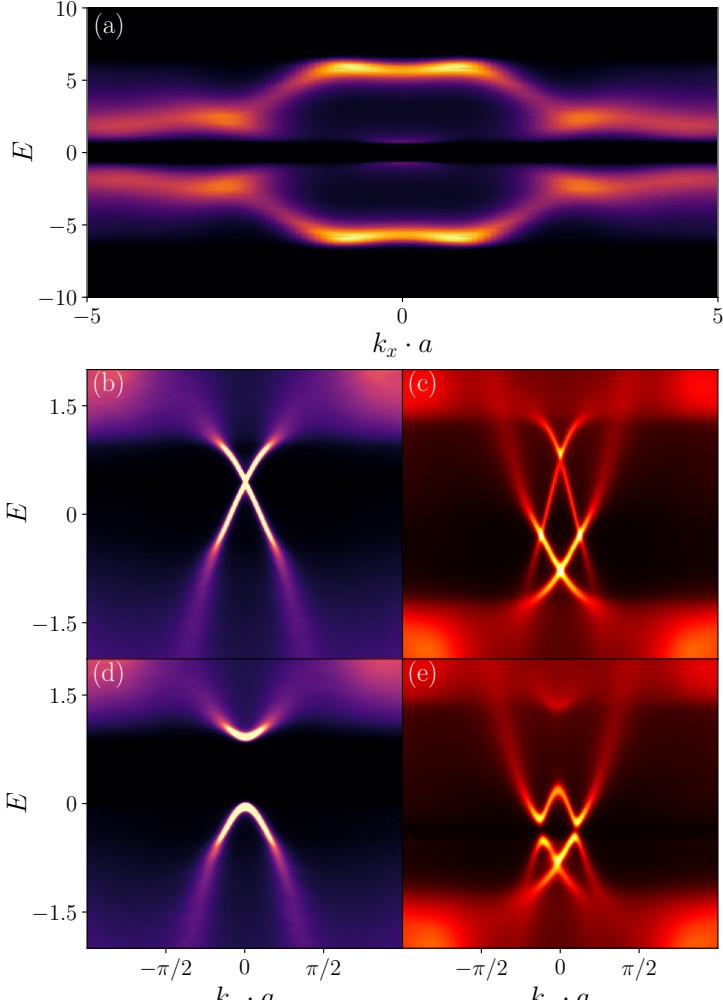

Figure 1:   The (a) bulk and (b)-(e) surface spectral functions of the amorphous tight-binding models. (b)-(c) The surface spectral functions of the $4 \times 4$ model (11) and the doubled $8 \times 8$ model (A.8). (d)-(e) the same models as (b)-(c) but with broken spatial (mirror and rotation) symmetries. Plot details are in App. B.

where the sums run over sites $\boldsymbol{r}$ and bonds $\langle \boldsymbol{r}, \boldsymbol{r}' \rangle$ in the system, and on-site (spin and orbital) degrees of freedom $i$ and $j$. The assumption of spatial isotropy and locality imposes symmetry constraints on the tight-binding terms [14]

$$U_g H^{\mathrm{hop}}(\boldsymbol{d}) U_g^{\dagger} = H^{\mathrm{hop}}(R_g \boldsymbol{d}), \tag{10}$$

for any symmetry group element $g$. This is also valid for onsite terms, treating them as hoppings with $\boldsymbol{d} = \boldsymbol{0}$. These constraints are formally identical to the constraints on the $\boldsymbol{k} \cdot \boldsymbol{p}$ models derived earlier, the only difference is that the hopping terms are generally nonhermitian, instead they obey the condition $H^{\mathrm{hop}}(\boldsymbol{d}) = H^{\mathrm{hop}}(-\boldsymbol{d})^{\dagger}$. Hence the $\boldsymbol{d}$-dependence of the symmetry-allowed hoppings has a similar structure to the $\boldsymbol{k}$-dependence of the symmetry-allowed $\boldsymbol{k} \cdot \boldsymbol{p}$ models. For the explicit form of the minimal tight-binding model obtained this way, see Appendix A. A compact form of the tight-

binding Hamiltonian is given by

$$H_{4\times4}^{\mathrm{onsite}} = \mu_1\sigma_0(\tau_0 + \tau_z)/2 + \mu_2\sigma_0(\tau_0 - \tau_z)/2, \tag{11}$$
$$H_{4\times4}^{\mathrm{hop}}(\boldsymbol{d}) = t_1(|\boldsymbol{d}|)\sigma_0(\tau_0 + \tau_z)/2 + t_2(|\boldsymbol{d}|)\sigma_0(\tau_0 - \tau_z)/2$$
$$+ t_3(|\boldsymbol{d}|)\boldsymbol{\sigma}\cdot\boldsymbol{d}\tau_+ + t_3^*(|\boldsymbol{d}|)\boldsymbol{\sigma}\cdot\boldsymbol{d}\tau_-,$$

where $t_1$ and $t_2$ are arbitrary real, and $t_3$ is an arbitrary complex functions of $|\boldsymbol{d}|$. If the complex phase of $t_3$ is constant, an on-site time-revesal symmetry of the form $\mathcal{T} = \sigma_y\exp(-i\tau_z\phi)\mathcal{K}$ where $\phi = \arg t_3$ would still exist, hence we demand a distance-dependent hopping phase in the spin-orbit hopping. We address question whether such a hopping term can arise in a realistic microscopic system in Sec. 2.3.

We examine the spectral functions of the minimal model, and confirm the joint presence of a spectral gap and the lack of spin splitting in the bulk [Fig. 1(a)], as expected from the symmetry analysis of the continuum model. The surface spectral function confirms the presence of gapless surface modes within the bulk gap [Fig. 1(b)]. For details see Sec. 3.2.

To further examine the extent of topological protection, we also define a model with twice the degrees of freedom and two Dirac cones on the surface in the continuum limit. We follow the same procedure as before, starting with two copies of the symmetry representation. This results in the $\boldsymbol{k}\cdot\boldsymbol{p}$ model $H_{8\times8}(\boldsymbol{k})$, and the associated tight-binding model $H_{8\times8}^{\mathrm{hop}}(\boldsymbol{d})$, which include generic coupling terms between the two copies, see Appendix A for details.

## 2.3 Microscopic implementation

Based on the symmetry-allowed terms of the amorphous tight-binding model (11), we now construct a microscopic hopping term that preserves isotropy while breaking TRS. The requirement to break TRS for the spin-orbit hopping connecting two orbitals with opposite inversion eigenvalues is that it has a distance-dependent phase in its amplitude. For simplicity, in the following we use the minimal model for a single bond connecting two different atoms that host spinful $s$ and $p_{x,y,z}$ orbitals respectively, as illustrated in Fig. 2(a). For the purpose of obtaining a minimal model, we separate the $p$ orbitals into $p_{3/2}$ and $p_{1/2}$ orbitals with an atomic spin-orbit coupling, and consider only the lower-energy $p_{1/2,\uparrow\downarrow}$ subspace.

In order to break TRS, we introduce magnetic atoms between the $s$ and $p$ orbitals, a plausible setup in an amorphous structure formed form chiral magnetic molecules. Hopping between the two atoms occurs through a virtual process via four $s$ orbitals on a plane perpendicular to the $s$–$p$ bond axis, located on the middle of the bond [Fig. 2(a)]. These intermediate $s$ orbitals each host a magnetic moment, such that together they form a chiral magnetic texture in the plane that contains them. The circulating magnetic texture defines a TRS-odd vector, that combined with the hopping vector $\boldsymbol{d}$, defines a scalar quantity $(\sum_n \boldsymbol{M}_n \times \boldsymbol{r}_n)\cdot\boldsymbol{d}$. This is the desired source of scalar TRS breaking. Such a magnetization configuration is also known in the literature as a toroidal moment [3,18], a time-reversal odd, polar vector order parameter $\boldsymbol{T} = \sum_n \boldsymbol{M}_n \times \boldsymbol{r}_n$ (the summation runs over the localized magnetic moments on the mid-bond plane). Tiling the space with such $s$–$p$ bonds restores spatial symmetries, while keeping TRS broken. The resulting structure can be viewed as collection of alternating sign magnetic toroidal monopoles [3,4]: the net toroidal moment of the bonds connected to each $s$ or $p$ site vanishes as $\sum_{\boldsymbol{d}} \boldsymbol{T}_{\boldsymbol{d}} \propto \sum_{\boldsymbol{d}} \boldsymbol{d} = 0$ (summing over all bonds $\boldsymbol{d}$ connected to a given site), which is also true on average for an isotropic amorphous structure. The localized toroidal monopole charge $T_0 = \sum_{\boldsymbol{d}} \boldsymbol{T}_{\boldsymbol{d}}\cdot\boldsymbol{d}$ is on the other hand nonzero, and takes opposite sign values on the $s$ and $p$ sublattices, providing a scalar, time-reversal odd order parameter. Such a magnetic texture may arise

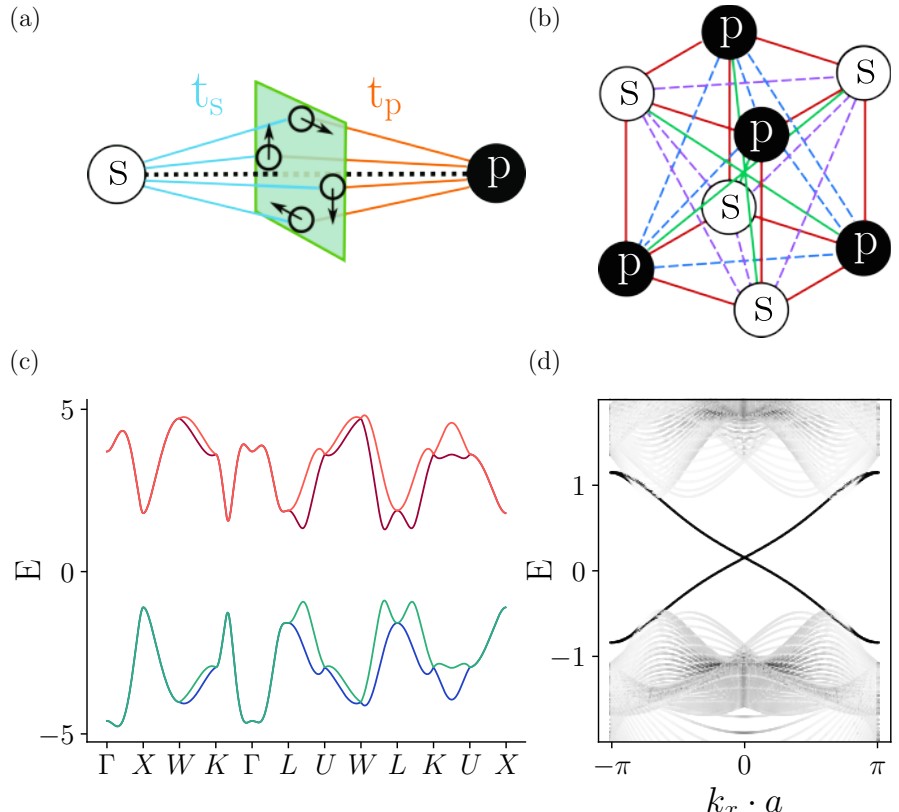

Figure 2:   Time-reversal symmetry breaking in a microscopic system with inversion and rotation symmetry. (a) A bond between $s$ and $p$ orbitals hosting four mid-bond $s$ orbitals (on plane shown in green) that host magnetic moments. (b) A section of a rock salt crystal structure made from the bond shown in (a). Red lines indicate nearest-neighbor hopping between $s$ and $p$ orbitals, dashed lines indicate second neighbor hopping between $s$ (purple) and $p$ (blue) orbitals, green lines indicate third neighbor hopping between $s$ and $p$ orbitals. (c) The bulk dispersion relation obtained from the crystal structure shown in (b) along the high-symmetry points of the face-centered cubic Brillouin zone. Different colors indicate different bands. (d) Bulk and surface dispersion of a 3D slab of the crystal. Darker color indicates a larger participation ratio. Plot details are in App. B.

in the presence of strong easy-axis anisotropy and Dzyaloshinskii–Moriya interaction between neighbouring magnetic moments on every bond. For a systematic breaking of TRS, all bonds must host the same toroidal moment, investigating the possible origin of such an ordered phase is beyond the scope of this manuscript.

The Hamiltonian of an $x$-aligned $s$–$p$ bond is:

$$H_m = E_s \sum_{\sigma} |s_\sigma\rangle \langle s_\sigma| + E_p \sum_{i,\sigma} |p_{i\sigma}\rangle \langle p_{i\sigma}| + \sum_{n,\sigma} (\Delta |s_{n\sigma}\rangle \langle s_{n\sigma}| + t_s |s_\sigma\rangle \langle s_{n\sigma}| + \text{h.c.})$$
$$+ \sum_{i,n,\sigma} (t_{in} |p_{i\sigma}\rangle \langle s_{n\sigma}| + \text{h.c.}) + \alpha \hat{\boldsymbol{L}}_p \cdot \hat{\boldsymbol{\sigma}}_p + \sum_{n} \boldsymbol{B}_n \cdot \hat{\boldsymbol{\sigma}}_n,$$

$$(12)$$

where $\sigma \in \{\uparrow, \downarrow\}$, $i \in \{x, y, z\}$, $n \in \{1, 2, 3, 4\}$, $|s_\sigma\rangle$ are the spinful $s$ orbital states, $|s_{n\sigma}\rangle$ are the mid-bond magnetic $s_n$ orbitals, $|p_{i\sigma}\rangle$ are the $p_{x,y,z}$ orbitals, $E_{s/p}$ are the onsite energies of the $s$ and $p$ orbitals, $\Delta$ is the onsite energy of the mid-bond $s_n$ orbitals, $\alpha$ is the magnitude of the atomic spin-orbit coupling splitting on the $p$ orbitals, $\hat{\boldsymbol{\sigma}}_{p/n}$ are the

spin operators on the $p$ and $s_n$ orbitals, $\hat{\boldsymbol{L}}_p$ are the orbital angular momentum operators on the $p$-orbitals, $\boldsymbol{B}_n$ are the magnetic moments of the $s_n$ orbitals. Finally, $t_{in}$ are the amplitudes of the $s_n - p_i$ hopping, determined by whether the hopping between the $p_{x,y,z}$ orbitals and the $s_n$ orbitals takes place via the positive or negative lobes of the $p$ orbitals:

$$t_{in} = t_x \delta_{ix} + t_{yz} \delta_{iy} \text{sgn}(y_n) + t_{yz} \delta_{iz} \text{sgn}(z_n) \tag{13}$$

where $y_n$ and $z_n$ are the $y$ and $z$ coordinates of the $s_n$ orbitals and $\text{sgn}(0) = 0$. In general the hopping parameters depend on the length of the bond, and because they come from overlap integrals of differently oriented $p$-orbitals, this dependence is different for $t_x$ and $t_{yz}$.

We use second-order quasi-degenerate perturbation theory (assisted by the Python software package Pymablock [19]) to obtain the effective hopping $t_{sp}$ between the $s$ and $p_{1/2}$ orbitals. We treat the decoupled atoms as the unperturbed Hamiltonian $H_0$, and all hopping terms as perturbations $H'$ with the $t$'s as small parameters. We divide the Hilbert-space into the low-energy subspace $A$ only containing the $s$ and $p_{1/2}$ subspaces on the two end atoms, and group all other degrees of freedom in the $B$ subspace. The second-order correction to the effective Hamiltonian for the $A$ subspace is given by [20]:

$$H^{(2)}_{mm'} = \frac{1}{2} \sum_{l \in B} H'_{ml} H'_{lm'} \left[ \frac{1}{E_m - E_l} + \frac{1}{E'_m - E_l} \right], \tag{14}$$

where $|m\rangle$ and $|l\rangle$ are orthonormal eigenstates of $H_0$ in the $A$ and $B$ subspaces respectively, and $E_l$ and $H'_{ml}$ denote eigenenergies and matrix elements in this basis. The zeroth order term in the effective Hamiltonian is given by the on-site energies, and the first-order term vanishes, so this is the only term of interest. In particular, we extract the off-diagonal block of the $4 \times 4$ effective Hamiltonian, and interpret it as the effective hopping matrix elements between the two end atoms.

We find that the resulting terms have the desired symmetries of the bond (rotations around the bond axis and mirrors including the bond axis) for arbitrary parameters. We demonstrate this result in a limiting case defined by the set of inequalities $\alpha \gg \Delta + B \gg \Delta - B \gg E_s,\ E_p - \alpha,\ t_s,\ t_{x/y/z}$, which holds when the atomic spin-orbit coupling and the local magnetic moments are large, so we only take into account hopping via the lower-energy virtual level $\Delta - B$. The resulting expression for the effective hopping is:

$$H^{\text{hop}}_{s-p} = \frac{t_s(2t_x - it_{yz})}{\sqrt{3}(\Delta - B)} i\sigma_x. \tag{15}$$

This hopping has a complex hopping phase, which breaks TRS. The hopping phase is distance dependent due to the different distance dependence of the microscopic hopping amplitudes from the $p_x$ and $p_{y,z}$ orbitals. This ensures that the hopping phase cannot be removed by a global basis-transformation introducing a relative phase between the $s$ and $p$ wavefunctions, resulting in an effective time-reversal symmetry. Hopping terms along directions other than $x$ follow from applying rotation operators, resulting in hopping terms of the form $H^{\text{hop}}_{s-p}(\boldsymbol{d}) = t_{sp}(|\boldsymbol{d}|)\, \boldsymbol{d} \cdot \boldsymbol{\sigma}$ where $\boldsymbol{d}$ is the hopping vector, and $t_{sp}$ is a complex function of the hopping distance given by the prefactor in (15). This has the same structure as the off-diagonal blocks of the hoppings in the minimal tight-binding model found in section 2.2 providing a proof-of-concept realization of the symmetry-allowed, scalar TRS-breaking hopping. For simplicity and without loss of generality, in the amorphous calculations we use the minimal tight-binding model with one type of atom with four degrees of freedom per atom, rather than a system with two families of atoms and two degrees of freedom per atom.

Before discussing the amorphous case, we demonstrate scalar TRS-breaking, by calculating the dispersion of a cubic rocksalt crystal endowed with this hopping term on

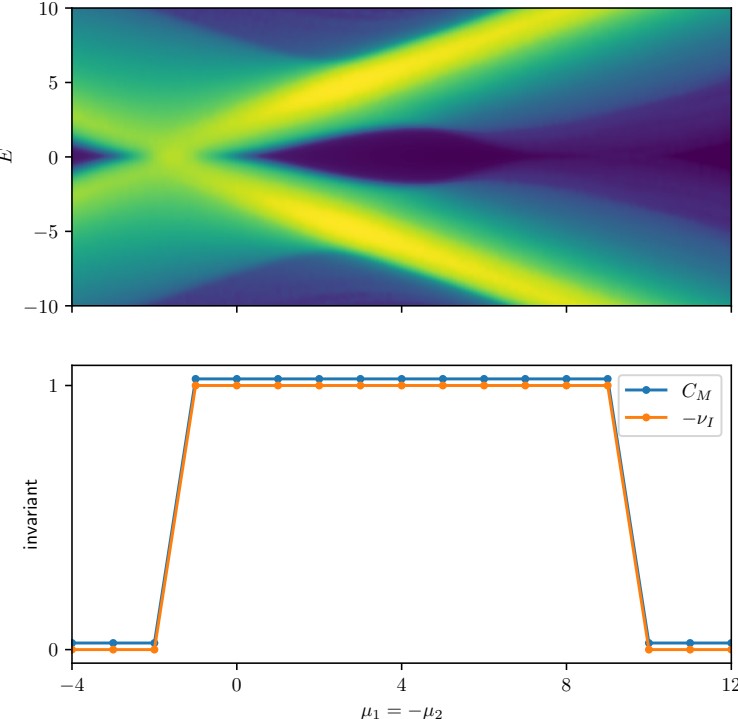

Figure 3: Topological phase transitions of the doubled class A amorphous tight-binding model (A.8) as a function of chemical potentials $\mu_1 = -\mu_2$, using parameters $t_3(d) = 1.2 \exp(-d) + i \exp(-0.3d)$, $t_1(d) = -2 \exp(-d)$, $t_2(d) = 2 \exp(-d)$. Top panel: Bulk density of states, showing gap closings and gapped phases as a function of the chemical potential. Brighter colors denote higher density of states in arbitrary units. Bottom panel: Topological invariants $C_M$ (defined in (17) and $\nu_I$ (E.1). Plots are offset for clarity.

the nearest neighbour and third neighbor $s$—$p$ bonds [Fig. 2(b), for details see App. D]. The dispersion relation shows that the bands are spin-split away from high-symmetry points and lines, demonstrating that TRS is systematically broken, while all space-group symmetries are preserved [Fig. 2(c)]. The surface dispersion shows gapless, propagating surface modes within the bulk gap, consistent with a crystalline mirror-Chern insulator state [Fig. 2(d)].

# 3  Topological properties

## 3.1  Bulk invariants

In order to establish connection between the continuum and tight-binding models, we use an effective $k$-space continuum Hamiltonian $H_{\text{eff}}$ that we obtain by inverting the single-particle Green's function that we project onto the plane wave basis, as described in Refs. [8, 14, 21, 22]:

$$\left(H_{\text{eff}}(\boldsymbol{k})^{-1}\right)_{lm} = G_{\text{eff}}(E, \boldsymbol{k})_{lm} = \langle \boldsymbol{k}, l | \hat{G}(E) | \boldsymbol{k}, m \rangle, \tag{16}$$

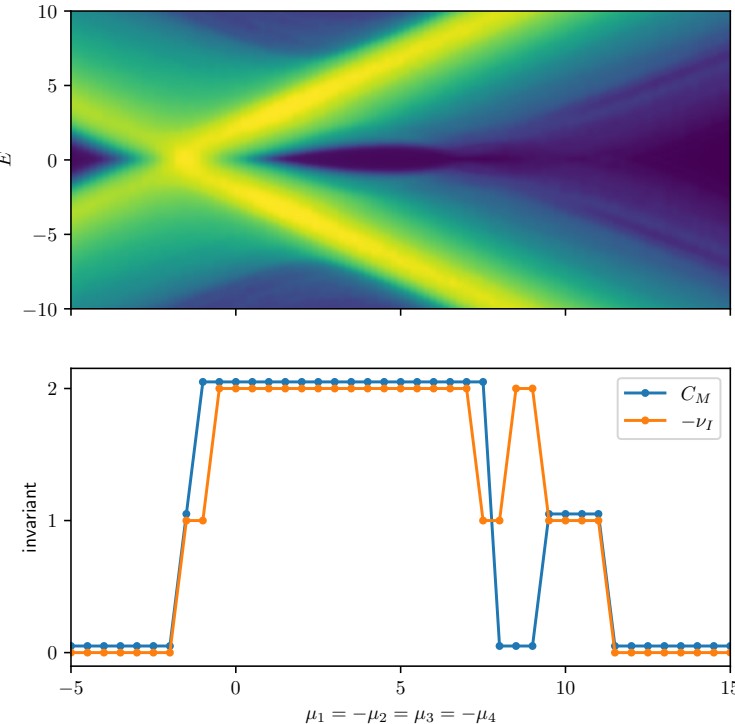

Figure 4:   Topological phase transitions of the doubled class A amorphous tight-binding model (A.8) as a function of chemical potentials $\mu_1 = -\mu_2 = \mu_3 = -\mu_4$, see Appendix B for the other parameters used. Top panel: Bulk density of states, showing several gap closings and gapped phases as a function of the chemical potential. Brighter colors denote higher density of states in arbitrary units. Bottom panel: Topological invariants $C_M$ and $\nu_I$. Plots are offset for clarity.

where $\hat{G}(E) = \left(E - \hat{H}\right)^{-1}$ is the Green's function of the full real-space Hamiltonian $\hat{H}$. The inverse is well defined if $E$ is in the spectral gap of $\hat{H}$, in the following we fix $E = 0$ and choose the chemical potentials, such that the gap is centered around zero energy. In the thermodynamic limit, due to self-averaging, $G_{\text{eff}}$ converges to the disorder-averaged Green's function. As a result, $G_{\text{eff}}$, as well as $H_{\text{eff}}$ inherit the average symmetries of the tight-binding model, and obey same symmetry constraints as the continuum models discussed in sec. 2.1. This also means, that the long-wavelength expansion of $H_{\text{eff}}(\boldsymbol{k})$ has the same functional form as the generic symmetry-allowed $\boldsymbol{k}\cdot\boldsymbol{p}$ model we found. In earlier work we argued that topological invariants of the compactified effective $\boldsymbol{k} \cdot \boldsymbol{p}$ models (extending $\boldsymbol{k}$-space with the point at infinity [14, 21]) provides at least a partial classification of the underlying amorphous systems, we will also follow this route here. We emphasize that the bulk invariants defined in the following rely on the possibility of defining an effective continuum Hamiltonian that is gapped and bounded for all momenta, which is not necessarily true if the self-energy has poles [23]. These invariants are valid and protected for continuum Hamiltonians with exact rotation symmetry, however, the classification might collapse in the presence of strong disorder, e.g. average rotation preserving dimerization.

The topological invariants of crystalline 3D mirror-Chern insulators are mirror Chern numbers, given by the difference in Chern numbers of opposite mirror sectors on mirror-

invariant 2D planes of the Brillouin-zone [5] In the presence of disorder, invariants of 3D statistical topological insulators are also constructed from the strong topological invariants of 2D subsystems [11]. Our 3D class A amorphous model relies on mirror symmetry to protect its surface modes, so motivated by the above results we find that the topological phase is characterized by a nontrivial value of the mirror-Chern invariant adapted to the amorphous continuum model:

$$C_M = \frac{1}{2}(C_+ - C_-), \quad C_\pm = \oiint \mathcal{F}_\pm(\boldsymbol{k})d^2\boldsymbol{k}, \tag{17}$$

where the integral runs over a compactified mirror-invariant plane $\mathbb{R}^2 \cup \{\infty\}$(e.g. $k_z = 0$, invariant under the mirror operator $k_z \to -k_z$ with $U_{M_z} = \mathcal{I}\exp(i\pi S_z)$), and $\mathcal{F}_\pm$ is the Berry curvature of the even/odd ($\pm i$ eigenvalue) mirror sub-blocks of the effective Hamiltonian. Because the system has inversion and rotation symmetries, the mirror Chern number can also be expressed in terms of rotation and inversion eigenvalues at high-symmetry momenta, $C_M = -\nu_I$, for details see App. E. We numerically evaluate both invariants for the $4 \times 4$ tight-binding model (11) using the parameters $\mu_2 = -\mu_1$, $t_3(d) = 1.2\exp(-d) + i\exp(-0.3d)$, $t_1 = -2$, $t_2 = 2$, on an amorphous sample with 14895 sites, the resulting topological phase transitions are shown in Fig. 3.

We also evaluated the invariants for the doubled model, see Fig. 4. We observe several topological phase transitions, with a large region corresponding to $C_M = 2$. The two invariants agree for most of the parameter range that we investigated, and we attribute the disagreement to the numerical instability of the mirror Chern number calculation in regions where the spectral gap is small.

## 3.2  Surface spectrum

As demonstrated in Fig. 2(d) for the crystalline system, the high-symmetry surface of the $C_M = 1$ model hosts a single Dirac cone, and multiple Dirac cones remain protected for $C_M > 1$. We expect that the high degree of ensemble averaged spatial symmetry of the amorphous Hamiltonian prevents surface states from being gapped out on any surface both for the single and doubled model ($C_M = 1$ and 2 respectively). We confirm this by numerically computing the surface spectral function

$$A(E, \boldsymbol{k}) = \sum_l \langle \boldsymbol{k}, l | \delta(\hat{H} - E) | \boldsymbol{k}, l \rangle, \tag{18}$$

using the Kernel polynomial method [24], specifically an implementation for computing the surface spectral functions in disordered systems [14, 25]. Here $\hat{H}$ is the real-space Hamiltonian of a finite slab, $l$ runs over the internal degrees of freedom, and $|\boldsymbol{k}, l\rangle$ is a plane-wave state localized on one surface.

We find that both the original and doubled amorphous models have a nonzero surface density of states in the bulk gap, with one or two Dirac nodes located at zero momentum. [Fig. 1(b,c)]. This is a consequence of the nontrivial topology of the continuum system described by the bulk effective Hamiltonian. The surface spectral function in the $k_x$ direction probes the topology of the $k_y = 0$ cut of the bulk effective Hamiltonian, which is invariant under $M_y$ in the thermodynamic limit. This allows decomposition into two mirror sectors, each of which is a Chern insulator, resulting in an edge spectrum with $C_M$ pairs of counter-propagating chiral edge states crossing the bulk gap. The modes with different chirality correspond to different mirror sectors, hence they are protected from gapping out by mirror-symmetric terms in the continuum model. We demonstrate that the surface states gap out when the symmetries protecting the topological phase

(rotations and mirrors normal to the surface) are broken on average [Fig. 1(d,e)]. It is not clear, however, whether disorder that respects the mirror symmetry on average is capable of opening a spectral gap in the amorphous system with even $C_M$. The transport calculations in the next section show that the surface of the even phase localizes, which suggests that a local surface perturbation compatible with the average symmetry is capable of opening a spectral gap.

## 3.3 Surface transport

Reference [11] conjectures that only the $\mathbb{Z}_2$ part of the invariant provides topological protection, or in other words, that only the surface states of systems with odd $C_M$ are protected from localization. In a crystalline system, the surface has an ensemble point group symmetry, and its localization properties are therefore equivalent to a doubled Chalker-Coddington network model, which has a localized phase with an anomalously large localization length [26, 27]. The conjecture, however, was not confirmed for 3D phases with continuous rotation symmetries, such as our amorphous model. To confirm the conjecture, we simulate the surface transport properties using amorphous network models.

We first simulate the transport properties of the regular network model as a baseline for the comparison. In the presence of disorder that preserves the spatial symmetries on average, the surface of the crystalline phase is equivalent to a critical Chern insulator. We simulate its transport properties with the Chalker-Coddington network model on the square lattice [28]. We fix the aspect ratio of the network to 1 and impose periodic boundary conditions along the $y$ direction [Fig. 5(a)]. The scattering matrices at each node of the network are random $2 \times 2$ matrices sampled from a Haar-distributed $U(2)$ ensemble. The conductance through the system is:

$$G = \frac{e^2}{h} \sum_i T_i,$$ (19)

where $T_i$ are the transmission probabilities from the modes entering one side of the network to the modes exiting on the other side. Since the aspect ratio equals to 1, the system conductivity $g = G$. We calculate the average conductivity $\langle g \rangle$ as a function of system size $L$ and reproduce the known result $\langle g \rangle \approx 0.5$–$0.6e^2/\hbar$ [29] [Fig. 5(d)], with the slow increase as a function of $L$ due to finite-size effects. We investigate the localization properties of the double Dirac cone model by doubling the number of modes on each link, as shown schematically in Fig. 5(c). This system is expected to localize, based on both numerical [26] and analytical [27] studies. We draw the $4 \times 4$ scattering matrices of the doubled networks from the circular unitary ensemble and confirm localization at system sizes of several thousand sites [Fig. 5(d)].

We now simulate the conductance of our amorphous model, in order to determine whether the average continuous rotation symmetry has an effect on the conductance properties of the system. We define an amorphous 2D network model in order to simulate the average rotation symmetry using a fourfold coordinated random graph [21,30], for details of the construction of the amorphous network see App. F. We use an annulus geometry in order to avoid issues constructing the network with periodic boundary conditions, and numerically calculate the conductance through the bulk from the modes entering the outer edge to the modes exiting the inner edge of the annulus [Fig. 5(b)]. The conductance $G$ is calculated using (19), and the conductivity of the annulus equals:

$$g = \frac{1}{2\pi} G \log \left( \frac{R}{r} \right),$$ (20)

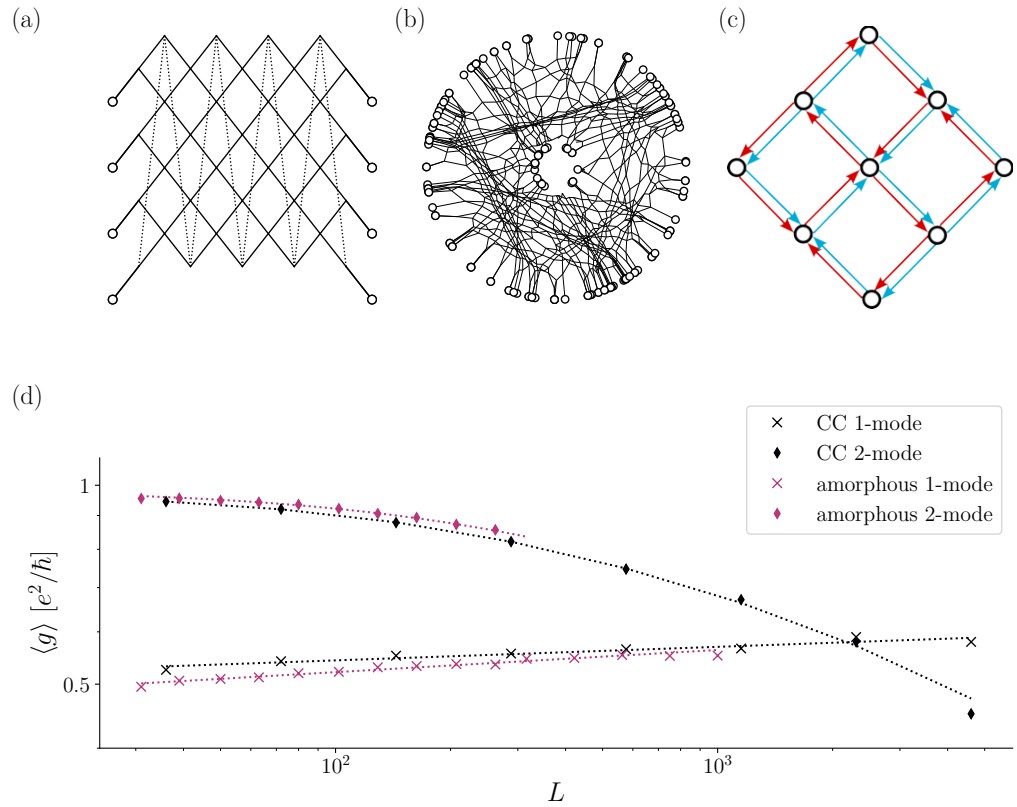

Figure 5:   Conductivity of translationally invariant and amorphous networks.   (a) Schematic of the Chalker-Coddington model. Dashed links loop in the vertical direction to indicate periodic boundary conditions. Circular nodes indicate external nodes where modes enter and exit the network. Internal nodes are located at all solid line crossings. (b) Schematic of the amorphous network. Circular nodes indicate external nodes where modes enter and exit the network. Nodes internal to the network are located at all line crossings. (c) Schematic of modes in the doubled model. (d) Average conductivity of the networks as a function of network length and width $L$ and fits (dashed lines). Results are shown for the Chalker-Coddington (CC) network and amorphous network, with 1 mode per link (crosses) and 2 modes per link (diamonds). Plot details are in App. B.

where $R$ and $r$ are the outer and inner radii of the annulus respectively. The results for the amorphous network closely follow the results for the regular network: the single Dirac cone conductivity falls within the $0.5 - 0.6 e^2/\hbar$ range for small $L$ and increases due to finite-size effects, and the double Dirac cone network localizes [Fig. 5(d)]. These observations confirm that a doubled phase transition is not protected from localization, even in the presence of average isotropy.

## 4   Discussion

In this work, we found that three-dimensional amorphous matter with average isotropy, but breaking all non-spatial symmetries host topologically protected phases of matter. We devised a rotation- and inversion-symmetric continuum model with broken time-reversal symmetry, and presented a microscopic realization of this model in amorphous matter with average isotropy. The feasibility of amorphous magnetic structures assembled from

chiral magnetic molecules or nanoparticles [31] is supported by experimental studies on Prussian blue analogues [32–34] and single-molecule magnets [35, 36], exhibiting magnetic interactions leading to ferrimagnetic and non-collinear antiferromagnetic ordering. We constructed a bulk $\mathbb{Z}$ invariant for the effective continuum model—expressible both in terms of symmetry eigenvalues and mirror Chern numbers— indicating the presence of a protected ungappable surface Dirac cone for odd values, which we numerically demonstrated.

We simulated the transport of our models using both regular and amorphous network models with random scattering at each node. We found critical conductance scaling for a single copy of the network (corresponding to the surface of a bulk with mirror Chern number $C_M = 1$), deviations from which are likely due to finite-size effects. Upon doubling the degrees of freedom in both the regular and amorphous networks, the modes localize as conjectured in Refs. [11, 26, 27]. Even though the numerics does not indicate a spectral gap forming for any higher number of surface Dirac cones, we expect that only an odd number are protected from localization and gapping out. We leave further investigation of the surface spectral properties in the even phases to future work.

Regardless of whether the surface states are protected, the question remains whether the bulk is topological for even values of the invariant. For example, in the case of topological crystalline phases protected by inversion symmetry only, it is known that clean systems do have bulk topological phases with fully gapped surfaces [37, 38]. Recent results show that in the presence of strong disorder the topological classification with average mirror symmetry collapses to $\mathbb{Z}_2$ [39], which supports our conclusion that the odd phase is topological, but suggests that the even phases might also be trivial in the bulk. We emphasize that the bulk invariants we define rely on the possibility of defining an effective continuum Hamiltonian (or equivalently, disorder-averaged Green's function) that is gapped and bounded for all momenta. This is not necessarily true if the self-energy diverges at certain momenta [23], which might be the case in the presence of strong disorder, e.g. dimerization. These considerations suggest that phases that differ by an even value of the bulk invariant may be topologically equivalent, but further work is needed to reach a definitive conclusion.

Due to the combination of average continuous rotation symmetry and inversion symmetry, the spin bands in the bulk of the amorphous system are doubly degenerate. This raises the question whether the systematic breaking of TRS leads to a macroscopic change in the material properties. Enumerating the possible non-dissipative electromagnetic responses compatible with isotropy and inversion-symmetry, but forbidden by TRS, we find $\boldsymbol{P} \propto \boldsymbol{E} \times \boldsymbol{B}$, electrical polarization parallel to the Poynting vector. This second-order response is distinct from the circular photogalvanic effect [40, 41], which only manifests in systems with broken inversion symmetry, and should therefore be absent in our system. The combination of these two responses therefore serve as a probe of the scalar TRS breaking.

A natural further question is, what is the classification of isotropic three-dimensional media with or without inversion symmetry in the other Altland-Zirnbauer symmetry classes [42]. The topological invariants outlined in this work remain valid if we also include TRS besides isotropy and inversion symmetry. Our models are compatible with prescribing TRS with the usual representation $\mathcal{T} = \exp(i\pi S_y)\mathcal{K}$, which fixes some parameters, but does not forbid any topological phases. In this case odd values of $C_M$ correspond to an amorphous strong topological insulator [43], however, the gapless surface Dirac cones remain protected by mirror symmetry for even values as well. To our knowledge, TRS does not enrich the classification in the presence of isotropy and inversion symmetry; and the classification with isotropy, broken inversion and unbroken TRS is the same as the

strong $\mathbb{Z}_2$ classification with TRS only. There is, however an interesting possibility that isotropy and the protection of the surface density of states in a doubled phase prevents the surface conductivity from going below the metal-insulator critical point, and because of that guaranteeing that the surface stays metallic. We leave an investigation of these properties to future work.

Our microscopic model—relying on orbital-selective hoppings through chiral magnetic molecules—demonstrates the difficulty of constructing a time-reversal odd, inversion even, scalar order parameter. In our case the order parameter is $\boldsymbol{P} \cdot (\boldsymbol{\nabla} \times \boldsymbol{M})$, electric polarization times bound current, which is equivalent to the toroidal magnetic monopole density [3,18]. Analyzing an effective field-theory displaying such order paramater without other symmetry breaking would shed further light on the properties of this class of isotropic magnetic materials.

# Data availability

The data shown in the figures is available at [44].

# Code availability

The code generating all of the data shown in the figures is available at [44].

# Author contributions

D. V. proposed the initial project idea, all authors contributed to creating the research plan and later refining it. D. V. formulated the bulk invariants. A. A. and D. V. devised the microscopic system and the scalar time-reversal breaking mechanism. D. V. wrote the code generating amorphous structures and computing the spectral functions. A. A. wrote the code for constructing and solving the network models. H. S. performed the numerical simulations and wrote the manuscript with input from all authors. A. A. managed the project with input from all authors.

# Acknowledgments

D. V. thanks Roderich Moessner for useful discussions. A. A. is grateful to Piet Brouwer for enlightening comments. The authors thank Elizabeth Dresselhaus and Bjorn Sbierski for sharing their network model code. The authors thank Isidora Araya Day for helping to set up and perform Pymablock calculations. A. A. and H. S. were supported by NWO VIDI grant 016.Vidi.189.180 and by the Netherlands Organization for Scientific Research (NWO/OCW) as part of the Frontiers of Nanoscience program. D. V. was supported by the Swedish Research Council (VR), the Knut and Alice Wallenberg Foundation, the Deutsche Forschungsgemeinschaft (DFG, German Research Foundation) under Germany's Excellence Strategy through the Würzburg-Dresden Cluster of Excellence on Complexity and Topology in Quantum Matter – ct.qmat (EXC 2147, project-id 57002544), and the National Research, Development and Innovation Office of Hungary under OTKA grant no. FK 146499.

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

# A  Model Hamiltonians

We use Qsymm to generate 3D class A models that respect inversion symmetry and isotropic continuous rotation symmetry, whose symmetry representations are:

$$U_{\mathcal{I}} = \sigma_0\tau_z, \ S_x = \frac{1}{2}\sigma_x\tau_0, \ S_y = \frac{1}{2}\sigma_y\tau_0, \ S_z = \frac{1}{2}\sigma_z\tau_0, \tag{A.1}$$

where $U_{\mathcal{I}}$ is the unitary part of the inversion operator, $S_{x,y,z}$ are the generators of continuous spin rotations around the $x$, $y$, and $z$ axes, and the unitary part of the corresponding rotation operator is given by $U = \exp(i\boldsymbol{n} \cdot \boldsymbol{S})$ with $\boldsymbol{n}$ the axis and angle of rotation, and $\tau$, $\sigma$ are the Pauli matrices. $\tau$ represents the orbital component, and $\sigma$ the spin component of the Hilbert space. The resulting model also has reflection symmetry on any 2D plane,

$$U_{\mathcal{M}_x} = i\sigma_x\tau_z, \ U_{\mathcal{M}_y} = i\sigma_y\tau_z, \ U_{\mathcal{M}_z} = i\sigma_z\tau_z, \tag{A.2}$$

where $U_{\mathcal{M}_{x,y,z}}$ is the unitary part of the reflection operators on the planes perpendicular to the $x$, $y$ and $z$ axes, or in general,

$$U_{\mathcal{M}_{\hat{\boldsymbol{n}}}} = \exp(i\pi\hat{\boldsymbol{n}} \cdot \boldsymbol{S})\tau_z, \tag{A.3}$$

where $\hat{\boldsymbol{n}}$ is a unit vector defining the mirror normal. Because of the full rotation invariance, prescribing one mirror symmetry results in mirror symmetry with respect to any plane.

The generated $k$-space model is listed in the main text in Eq. (7). In real-space, the model is of the form:

$$H_{4\times4}^{\text{onsite}} = \mu_1\sigma_0(\tau_0 + \tau_z)/2 + \mu_2\sigma_0(\tau_0 - \tau_z)/2, \tag{A.4}$$

$$\begin{aligned} H_{4\times4}^{\text{hop}}(\boldsymbol{d}) &= (tn_1 + t_2d^2)\sigma_0(\tau_0 + \tau_z)/2 + (tn_2 + t_3d^2)\sigma_0(\tau_0 - \tau_z)/2 \\ &\quad + (t_0 - t_5d^2)\boldsymbol{\sigma} \cdot \boldsymbol{d}\tau_y + (t_1 + t_4d^2)\boldsymbol{\sigma} \cdot \boldsymbol{d}\tau_x, \end{aligned} \tag{A.5}$$

where $tn_i$ are normal hopping terms, $\boldsymbol{d} = (d_x, d_y, d_z)$, with $d_i$ the bond lengths along axis $i \in \{x, y, z\}$ that connect neighboring sites, and $d^2 = \boldsymbol{d} \cdot \boldsymbol{d}$. In general, the symmetry is still preserved if the hopping parameters have arbitrary dependence on the bond length $d$, so in certain calculations we set $t_4 = t_5 = 0$ and make $t_0$ and $t_1$ depend exponentially on the

bond length with different scaling, see Appendix B. When demonstrating that symmetry-breaking gaps out the surface Dirac-nodes, we introduce a mass term that breaks all symmetries except for continuous rotation around the $x$ axis:

$$\lambda = (\sigma_0 + \sigma_x)\tau_y. \tag{A.6}$$

We also construct a doubled model by doubling the number of degrees of freedom. The new symmetry representation is two copies of the one above, obtained by the replacement $U_g \rightarrow \rho_0 \otimes U_g$, where $\rho_0$ is the $2 \times 2$ identity matrix acting on the space of the two copies. The generic symmetry-allowed $\boldsymbol{k} \cdot \boldsymbol{p}$ model takes the form:

$$
\begin{aligned}
H_{8\times8}(\boldsymbol{k}) ={}& 1/2(\rho_0 + \rho_z)\sigma_0(\mu_1(\tau_0 + \tau_z)/2 + \mu_2(\tau_0 - \tau_z)/2) \\
& + 1/2(\rho_0 - \rho_z)\sigma_0(\mu_3(\tau_0 + \tau_z)/2 + \mu_4(\tau_0 - \tau_z)/2) \\
& + \rho_+\sigma_0(\lambda_1(\tau_0 + \tau_z)/2 + \lambda_2(\tau_0 - \tau_z)/2) \\
& + \rho_-\sigma_0(\lambda_3(\tau_0 + \tau_z)/2 + \lambda_4(\tau_0 - \tau_z)/2) \\
& + (t_0(\rho_0 + \rho_z)/2 + t_3(\rho_0 - \rho_z)/2)\boldsymbol{\sigma} \cdot \boldsymbol{k}\tau_x \\
& - (t_4(\rho_0 + \rho_z)/2 + t_7(\rho_0 - \rho_z)/2)\boldsymbol{\sigma} \cdot \boldsymbol{k}\tau_y \\
& + (t_1 + it_5)\rho_-\boldsymbol{\sigma} \cdot \boldsymbol{k}\tau_- + (t_1 - it_5)\rho_+\boldsymbol{\sigma} \cdot \boldsymbol{k}\tau_+ \\
& + (t_2 + it_6)\rho_-\boldsymbol{\sigma} \cdot \boldsymbol{k}\tau_+ + (t_2 - it_6)\rho_+\boldsymbol{\sigma} \cdot \boldsymbol{k}\tau_-
\end{aligned} \tag{A.7}
$$

where $\mu_i$ are chemical potential terms, $\lambda_i$ are symmetry-allowed onsite mixing between states with the same symmetry character in the two copies, $t_i$ are the hopping terms, $\rho$, $\sigma$ and $\tau$ are the Pauli matrices, $\boldsymbol{k} = (k_x, k_y, k_z)$, and $k^2 = \boldsymbol{k} \cdot \boldsymbol{k}$. The onsite terms that are off-diagonal in $\rho$ can be removed by a symmetry-preserving basis transformation without changing the structure of the $\boldsymbol{k}$-dependent terms which couple the two copies, however, we include them in the numerics for generality. We do not include higher-order terms, because this $\boldsymbol{k}$-linear model already breaks all other symmetries.

In real space, the model takes the form:

$$
\begin{aligned}
H_{8\times8}^{\text{onsite}} ={}& 1/2(\rho_0 + \rho_z)\sigma_0(\mu_1(\tau_0 + \tau_z)/2 + \mu_2(\tau_0 - \tau_z)/2), \\
& + 1/2(\rho_0 - \rho_z)\sigma_0(\mu_3(\tau_0 + \tau_z)/2 + \mu_4(\tau_0 - \tau_z)/2) \\
& + \rho_+\sigma_0(\lambda_1(\tau_0 + \tau_z)/2 + \lambda_2(\tau_0 - \tau_z)/2) \\
& + \rho_-\sigma_0(\lambda_3(\tau_0 + \tau_z)/2 + \lambda_4(\tau_0 - \tau_z)/2) \\
H_{8\times8}^{\text{hop}}(\boldsymbol{d}) ={}& 1/2(\rho_0 + \rho_z)\sigma_0(tn_1(\tau_0 + \tau_z)/2 + tn_2(\tau_0 - \tau_z)/2) \\
& + 1/2(\rho_0 - \rho_z)\sigma_0(tn_3(\tau_0 + \tau_z)/2 + tn_4(\tau_0 - \tau_z)/2) \\
& + (it_0(\rho_0 + \rho_z)/2 + it_3(\rho_0 - \rho_z)/2)\boldsymbol{\sigma} \cdot \boldsymbol{d}\tau_x \\
& - (it_4(\rho_0 + \rho_z)/2 + it_7(\rho_0 - \rho_z)/2)\boldsymbol{\sigma} \cdot \boldsymbol{d}\tau_y \\
& + (-t_5 + it_2)\rho_-\boldsymbol{\sigma} \cdot \boldsymbol{d}\tau_- + (t_5 + it_2)\rho_+\boldsymbol{\sigma} \cdot \boldsymbol{d}\tau_+ \\
& + (-t_6 + it_1)\rho_-\boldsymbol{\sigma} \cdot \boldsymbol{d}\tau_+ + (t_2 + it_6)\rho_+\boldsymbol{\sigma} \cdot \boldsymbol{d}\tau_-,
\end{aligned} \tag{A.8}
$$

where $tn_i$ are normal hopping terms, $\boldsymbol{d} = (d_x, d_y, d_z)$, with $d_i$ the bond lengths along axis $i \in \{x, y, z\}$ that connect neighboring sites, and $d^2 = \boldsymbol{d} \cdot \boldsymbol{d}$.

When examining the effect of explicit symmetry-breaking in the doubled model, we use the term

$$\lambda' = \begin{pmatrix} 1 & 1 \\ 1 & 1 \end{pmatrix} \otimes \begin{pmatrix} 1 & 1 \\ 1 & 1 \end{pmatrix} \otimes \tau_y. \tag{A.9}$$

# B   Model and plotting parameters

In this section additional details of the plots are listed in order of appearance.

For panel (c) of Fig. 2 the Hamiltonian (D.1) was simulated using kwant [45] on a translationally invariant 3D face-centered cubic (FCC) lattice. Its eigenvalues were obtained along the high-symmetry points of the FCC lattice, using the parameters $\mu_1 = 0.1$, $\mu_2 = 0.2$, $t_1 = 0.3$, $t_2 = -0.4$, $t_3 = \exp(0.3i)$, $t_4 = 0.2i \exp(0.3i)$. For the dispersion shown in panel (d), a slab was simulated, periodic along the vectors $[1, 0, 0]$ and $[0, 1, 0]$, and with a width of 20 sites in the $[0, 0, 1]$ direction. The parameters used are the same as for panel (c).

For panel (a) of Fig. 5, the Chalker-Coddington network is composed of four unit cells in both $x$ and $y$. For panel (b), the amorphous network was created with an outer radius of $R = 20$, an inner radius of $r = 4$, and a density of 1. The positions of the nodes of the network underwent a relaxation step where the position of each node is sequentially averaged over the position of all neighboring nodes. For panel (d), the results for single-mode Chalker-Coddington network were obtained for 249 different random scattering matrix configurations, for network sizes of 36, 72, 144, 288, 576, 1152, 2304 and 4608 unit cells, with an aspect ratio of 1. The results for the two-mode Chalker-Coddington network were obtained for the same network sizes and aspect ratio, and for 269 different scattering matrix configurations. For the amorphous network, the results were obtained for 50 outer radii sizes between $10^{1.5}$ and $10^{2.5}$, with a fixed outer radius over inner radius ratio of 1.5, and a density of 0.7. Results for the single mode network were obtained for 500 different amorphous network and scattering matrix configurations, and 300 different configurations for the two-mode amorphous network. Additional results for the single mode network were obtained for 5 outer radii sizes between $10^{2.5}$ and $10^3$, for 100 different network configurations and scattering matrices.

For Fig. 1(a), single-Dirac cone model as defined in Eq. (A.4) was used. Its parameters were set to $\mu_1 = -1$, $\mu_2 = 1$, $tn_1 = 0$, $tn_2 = 0$, $t_0 = 0.5$, $t_1 = 0.4$, $t_2 = 1$, $t_3 = -1$, $t_4 = 0.3$, $t_5 = 0.8$, $\lambda = 0$. For panels (b) and (d) the same model as panel (a) was used with parameters $\mu_1 = 1$, $\mu_2 = -1$, $tn_1 = -2$, $tn_2 = 2$, $t_0 = 1$, $t_1 = 1$, $t_2 = 1.1$, $t_3 = 1.2$, $t_4 = 1.3$, $t_5 = 1.25$. The amorphous slab was generated in a box of dimensions $200 \times 50 \times 50$ and density 0.4. For panels (a) and (b) the additional symmetry-breaking term $\lambda$ from Eq. (A.6) is set to 0, and in panel (d), $\lambda = 0.3$.

For the doubled model shown in Fig. 1 panels (c) and (e) and Fig. 4 as defined in Eq. (A.8), the parameters were set to $\mu_1 = 1$, $\mu_2 = -1$, $\mu_3 = 1$, $\mu_4 = -1$, $tn_1 = -2$, $tn_2 = 2$, $tn_3 = -2$, $tn_4 = 2$, $t_i = 0.9$, $\lambda_1 = 0.1$, $\lambda_2 = 0.11$, $\lambda_3 = 0.12$, $\lambda_4 = 0.123$, and all hopping terms are multiplied by a distance-dependent factor $\exp(-d)$, except for $t_0$ where the factor $\exp(-0.3d)$ was used to achieve a distance-dependent hopping phase. For Fig. 1 The amorphous slab was generated in a box of dimensions $200 \times 50 \times 50$ and density 0.4. For panel (c) the additional symmetry-breaking term $\lambda'$ from Eq. (A.9) is set to 0, and in panel (e), $\lambda' = 0.3$.

For Fig. 3, the model (A.4) was used. For all results, the hopping parameters were set to $t_0 = 1$, $t_1 = 1.2$, $t_2 = 0$, $t_3 = 0$, $t_4 = 0$, $t_5 = 0$, $tn_1 = -2$, $tn_2 = 2$ (terms proportional to $d$ to the power of 2 and higher are set to 0). Since the only hopping terms are linear in $d$, in order to ensure that TRS is broken for this model, a different distance dependence is given for the $t_0$ and $t_1$: $t_0 \exp(-0.3d)$ and $t_1 \exp(-d)$, where $d = \sqrt{d^2}$ is the bond length. The amorphous samples are all contained within a cube of $30 \times 30 \times 30$ sites, with a density of 0.5. For the invariant $C_M$ (17) the numerical integration of the Berry curvature over the $\mathbf{k}$-space sphere was done over a grid of $10 \times 10$ points.

For Fig. 4, we used the same doubled model and parameters as for Fig. 1 panels (c)

and (d). The amorphous samples are all contained within a cube of $20 \times 20 \times 20$ sites, with a density of 0.5. For the invariant $C_M$ (17) the numerical integration of the Berry curvature over the $\boldsymbol{k}$-space sphere was done over a grid of $10 \times 10$ points.

For panel (b) of Fig. E.1, the model (D.1) was used. The parameters were set to $t_1 = 0.3$, $t_2 = -0.4$, $t_3 = \exp(0.3i)$, $t_4 = i\exp(0.3i)$. The $\Gamma$ and $X$ points of the model are $(0, 0, 0)$ and $(0, 2\pi, 0)$.

# C   Isotropy of the amorphous model

In this appendix we confirm that the amorphous tight-binding model produces an isotropic electronic structure up to random fluctuations. The underlying amorphous structure was obtained by the same method as in Ref. [46], where we also confirmed the isotropy of its two-point correlation function, hence here we focus only on the isotropy of the electronic spectral function.

We generated amorphous structures in a box of dimensions $50 \times 50 \times 50$ with density 0.4, and calculated the spectral function by sampling a ball of radius 20 in the middle of the sample, with an average $N = 13400$ lattice sites. Same as Fig. 1(a), single-Dirac cone model as defined in Eq. (A.4) was used with parameters set to $\mu_1 = -1$, $\mu_2 = 1$, $tn_1 = 0$, $tn_2 = 0$, $t_0 = 0.5$, $t_1 = 0.4$, $t_2 = 1$, $t_3 = -1$, $t_4 = 0.3$, $t_5 = 0.8$. For the rest of this analysis, we fixed $|\mathbf{k}| = 1$ in inverse length units, and took 500 samples for the spectral function $A(E, |\mathbf{k}| = 1)$ (with $E$ sampled at 400 values) from the following three random ensembles:

- Fixed disorder realization, random $\mathbf{k}$ with $|\mathbf{k}| = 1$,

- Random disorder realization, fixed $\mathbf{k} = (0, 0, 1)$,

- Random disorder realization, random $\mathbf{k}$ with $|\mathbf{k}| = 1$.

We plot the resulting distributions of $A(E, |\mathbf{k}| = 1)$ in Fig. C.1 top panel. The expectation values of the three distributions are indistinguishable, as illustrated in Fig. C.1 middle panel. When comparing the standard deviations, we find that the case with fixed disorder realization has significantly lower variance, while the other two are very similar, see Fig. C.1 bottom panel. It is is expected that a fixed disorder realization results in lower variance, as the samples from nearby $\mathbf{k}$-points are correlated, as illustrated in Fig. C.2. The relative fluctuation of the spectral function amplitude is in the range of $1-2\%$ in all cases, a value expected from statistical fluctuations in a finite sample of this size, scaling with $\sqrt{N}$.

We further compare the two cases with random disorder realizations, by calculating the statistical $p$-value and the Kolmogorov–Smirnov statistic $D$ for every $E$, see Fig. C.3. These both measure the similarity of the random distributions given a finite sample, high $p$ values and low $D$ values indicate high similarity, with $p = 1$ and $D = 0$ corresponding to identical samples. We find that at most $E$ values the distributions are sufficiently similar, and there are only a few outliers where we should reject the null hypothesis that the underlying distributions are identical with 95% confidence. Such outliers are, however, expected to occur in a set of 400 random distributions. Hence we conclude, that the electronic structures obtained in our numerics are isotropic up to random fluctuations.

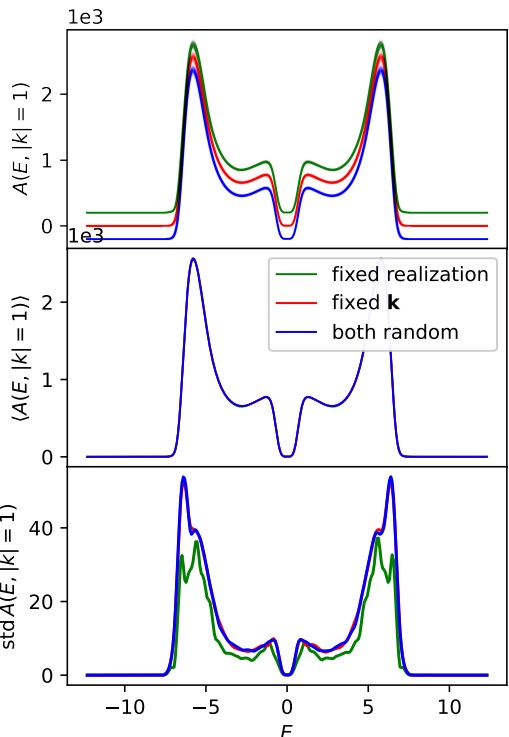

Figure C.1:  Spectral function statistical properties of the amorphous model. Top panel: Probability density of the spectral function as the function of energy $E$ at $|\mathbf{k}| = 1$ for three different ensembles. The plots are offset for visibility, and more saturated colors denote higher probability density. Middle panel: Expectation value of the spectral function. The three graphs completely overlap at this scale. Bottom panel: Standard deviation of the probability distributions.

# D    Spin splitting in a crystal

Because the scalar TRS breaking is insufficient to cause a spin splitting in an isotropic medium, we demonstrate the spin splitting in a crystal structure. We use the $s$ and $p$ atoms as the basis of the rock salt crystal structure [Fig. 2(b)] with full cubic ($O_h$) symmetry. In this model, orbitals of the same type are connected by normal hopping, and orbitals of different types are connected by the complex spin-orbit hopping of (15), resulting in terms off-diagonal in the orbital ($\tau$) space. Because the symmetry-breaking mechanism relies on the nontrivial distance-dependence of the hopping phase, we include both nearest-neighbor as well as third neighbor $s$–$p$ hopping [Fig. 2(b)]. We emphasize that this is a minimal model used as a sanity-check, hence we ignore the problem with microscopic realization posed by the third-nearest-neighbour bonds crossing each other.

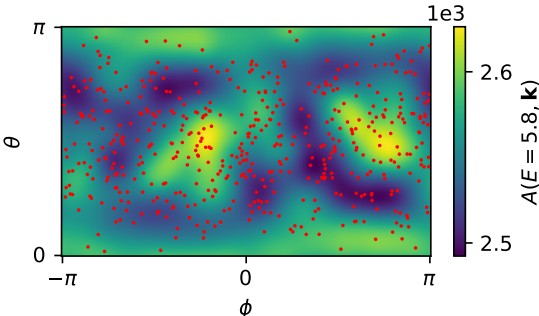

Figure C.2: Spectral function amplitude at $E = 5.8$ and $|\mathbf{k}| = 1$ for a fixed disorder realization as a function of the polar angles of $\mathbf{k}$. The results are interpolated, the red dots mark the sampled points.

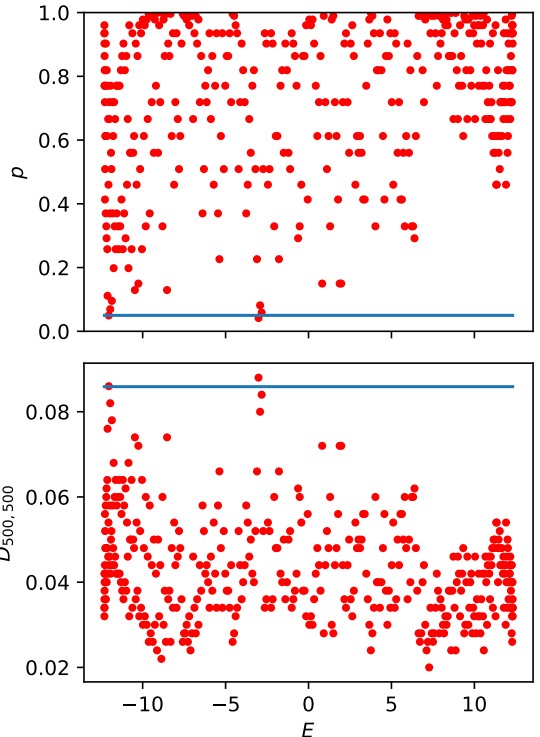

Figure C.3: Comparison of the probability distributions with random disorder realization with fixed or random $\mathbf{k}$. Top panel: statistical $p$ values. The blue line denotes $p = 0.05$, for lower $p$ the null hypothesis of the distributions being identical is rejected with 95% confidence. Bottom panel: Two-sample Kolmogorov–Smirnov statistic with 500 samples each. The blue line denotes the value over which the null hypothesis is rejected with 95% confidence.

The tight-binding Hamiltonian thus takes the form:

$$
\begin{aligned}
H_{\text{salt}} = {} & \left( \mu_1 + t_1 \sum_{\boldsymbol{d_2}} e^{i\boldsymbol{k}\cdot\boldsymbol{d_2}} \right) \sigma_0(\tau_0 + \tau_z)/2 + \left( \mu_2 + t_2 \sum_{\boldsymbol{d_2}} e^{i\boldsymbol{k}\cdot\boldsymbol{d_2}} \right) \sigma_0(\tau_0 - \tau_z)/2 \\
& + \frac{i}{a} \left( \sum_{\boldsymbol{d_1}} e^{i\boldsymbol{k}\cdot\boldsymbol{d_1}} \boldsymbol{d_1} \cdot \boldsymbol{\sigma} \right) (t_3\tau_+ + t_3^*\tau_-) + \frac{i}{a} \left( \sum_{\boldsymbol{d_3}} e^{i\boldsymbol{k}\cdot\boldsymbol{d_3}} \boldsymbol{d_3} \cdot \boldsymbol{\sigma} \right) (t_4\tau_+ + t_4^*\tau_-) ,
\end{aligned}
$$

$$(D.1)$$

where $a$ is the cubic cell lattice constant, $\sigma_\pm = \frac{1}{2}(\sigma_x \pm i\sigma_y)$, and similarly for $\tau_\pm$. $\boldsymbol{d}_1$ runs over the six nearest-neighbor bonds symmetry-equivalent to $\frac{a}{2}(1,0,0)$, $\boldsymbol{d}_2$ over the twelve second neighbor bonds symmetry-equivalent to $\frac{a}{2}(1,1,0)$, and $\boldsymbol{d}_3$ over the eight third neighbor bonds symmetry-equivalent to $\frac{a}{2}(1,1,1)$. The terms of Eq. (D.1) proportional to $t_1$ and $t_2$ are the second neighbor $s-s$ and $p-p$ normal hoppings respectively [dashed lines of Fig. 2(b)], where $t_1$ and $t_2$ are both real. The terms proportional to $t_3$ and $t_4$ are the nearest and third neighbor $s$–$p$ hoppings respectively [solid lines of Fig. 2(b)], with $t_3$ and $t_4$ complex. This Bloch Hamiltonian reproduces the symmetry-allowed terms of the continuum model (7) in the long-wavelength limit up to third order in $k$, aside from an additional cubic anisotropy term and a slight change of parametrization.

The tight-binding model (D.1) preserves the space group of the rock salt crystal structure [see App. A]. The spin-orbit-like $s$–$p$ hopping terms alternate in sign along the hopping axes in order to preserve inversion symmetry. We select the parameters $\mu_1 = 0.1$, $\mu_2 = 0.2$, $t_1 = 0.3$, $t_2 = -0.4$, $t_3 = \exp(0.3i)$, $t_4 = 0.2i\exp(0.3i)$. The dispersion relation shows that the spin bands are split away from high-symmetry points and lines that have at least a rotation and a mirror symmetry, demonstrating that TRS is broken [Fig. 2(c)]. The TRS-breaking is a result of the different $k$-dependence of the first and third neighbor hopping terms after a series expansion around $\boldsymbol{k} = 0$. A generic choice of the complex hopping amplitudes $t_3$ and $t_4$ leads to a $k$-dependent phase in the Bloch Hamiltonian, preventing the existence of a $k$-dependent effective TRS operator discussed in Sec. 2.1. The surface dispersion shows gapless, propagating surface modes within the bulk gap [Fig. 2(d)].

# E   Alternative bulk invariants

In addition to the bulk invariant given in Sec. 3.1, we identify two alternative expressions.

## E.1   Inversion eigenvalues

The inversion operator commutes with the spins at the rotation-invariant points $\boldsymbol{k} = \boldsymbol{0}$ and $\boldsymbol{k} = \infty$. Since the SU(2) rotation symmetry commutes with the inversion operator, the inversion eigenvalues come in degenerate pairs in the case of a spin-1/2 representation, and in degenerate groups of $2s+1$ for spin-$s$ representations. The difference in parity of the inversion eigenvalue pairs at these rotation-invariant points characterizes the topological phase:

$$\nu_I = \frac{1}{2}\left[\iota_-(\infty) - \iota_-(\boldsymbol{0})\right], \tag{E.1}$$

$$\iota_-(\boldsymbol{k}) = \mu_{-1}\left(\langle n(\boldsymbol{k})|\,\mathcal{I}\,|m(\boldsymbol{k})\rangle\right),$$

where $|n(\boldsymbol{k})\rangle$ are the occupied states of the effective Hamiltonian $H_{\text{eff}}$, and $\mu_\lambda(A)$ indicates the multiplicity of the eigenvalue $\lambda$ in the spectrum of $A$. We note that in the case of an operator that only has $\pm 1$ eigenvalues, the multiplicity can be expressed through the trace as $\operatorname{Tr} A = N - 2\mu_{-1}(A)$, allowing to rewrite the invariant as

$$\nu_I = -\frac{1}{4}\sum_{n\in\text{occ}}\left(\langle n(\infty)|\,\mathcal{I}\,|n(\infty)\rangle - \langle n(\boldsymbol{0})|\,\mathcal{I}\,|n(\boldsymbol{0})\rangle\right), \tag{E.2}$$

where we used that the total number of occupied bands is the same at $\boldsymbol{k} = \boldsymbol{0}$ and $\infty$.

While we only consider spin-1/2 representations in the main text, in the general case it is possible to resolve the eigenstates at $\boldsymbol{k} = \boldsymbol{0}$ and $\infty$ based on the spin-representation

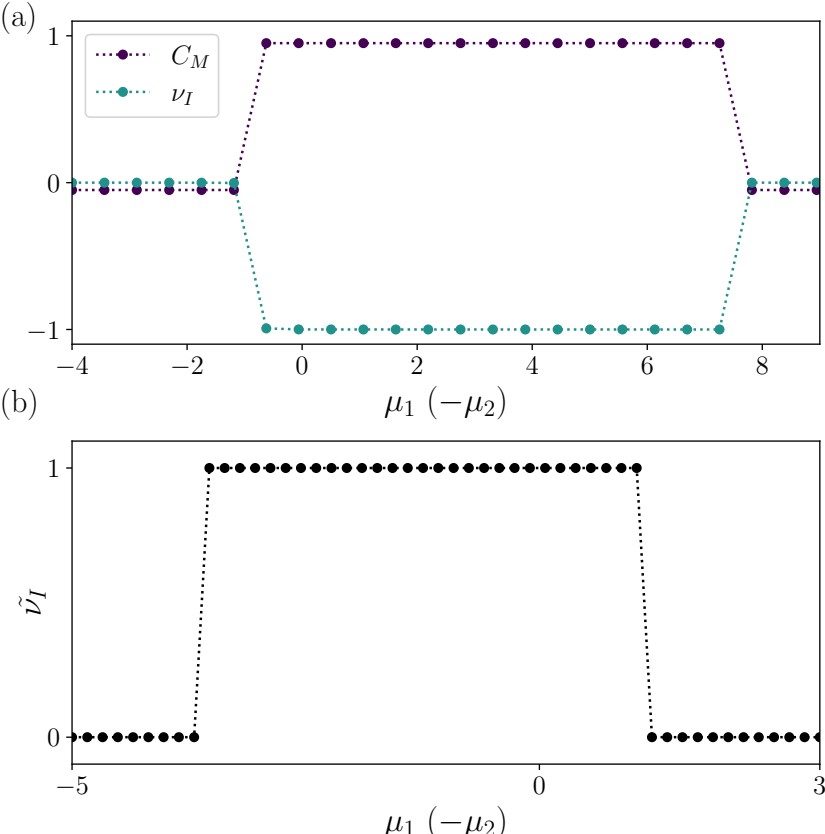

Figure E.1: (a) The topological invariants of the class A model (A.4) for amorphous systems ($C_M$ defined in (17) and $\nu_I$ in (E.1)) as a function of chemical potentials $\mu_{1,2}$. Plots are offset for clarity. (b) The invariant $\tilde{\nu}_I$ of the crystal system as a function of chemical potentials $\mu_{1,2}$ (D.1). Plot details are in App. B.

**$S$.** All states along a line $\hat{\boldsymbol{n}}k$ connecting $\boldsymbol{0}$ and $\boldsymbol{\infty}$ have continuous rotation symmetry along the $\hat{\boldsymbol{n}}$ axis, hence the eigenvalues of $\hat{\boldsymbol{n}} \cdot \boldsymbol{S}$ in the occupied subspace are well-defined throughout, and the total number of various spin representations cannot change. The inversion eigenvalues, however, can change in the process, so we can define the set of invariants

$$\nu_I^s = \frac{1}{2s+1} \left[ \iota_-^s(\boldsymbol{\infty}) - \iota_-^s(\boldsymbol{0}) \right], \tag{E.3}$$
$$\iota_-^s(\boldsymbol{k}) = \mu_{-1} \left( \langle n_s(\boldsymbol{k})| \, \mathcal{I} \, |m_s(\boldsymbol{k})\rangle \right),$$

where we restrict the inversion operator to the subspace corresponding to the spin-$s$ representation spanned by the states $|n_s(\boldsymbol{k})\rangle$. This results in a $\mathbb{Z}^{\mathbb{N}}$ classification, of which the invariant (E.1) only probes a $\mathbb{Z}$ subset,

$$\nu_I = \sum_s \left( s + \frac{1}{2} \right) \nu_I^s. \tag{E.4}$$

This relation also shows that, depending on the spin representation content of the model, not all values of $\nu_I$ may be realizable. A remaining question is, whether for general $s$, $\nu_I$ or the set of $\nu_I^s$ has a bulk-boundary correspondence in amorphous systems. As we show in the next section (see (E.9)), it is a different combination of $\nu_I^s$ that the mirror Chern invariant probes, nontrivial values of which we expect to protect robust surface states. The

simplest continuum model with trivial $\nu_I$ (or $C_M$) and nontrivial $\nu_I^s$ has 16 on-site degrees of freedom (4 spin-1/2 and 2 spin-3/2 representations, half of which is inversion-odd), we leave analysis of the surface physics to future work.

For the crystalline system described in Sec. D we calculate the analogous eigenvalue parity invariant given by:

$$\tilde{\nu}_I = \frac{1}{2} \left[ \iota_-(\Gamma) + \iota_-(X) \right] \mod 4, \tag{E.5}$$

where $\iota$ is the same as in (E.1). The mod 4 results from factoring out atomic insulators located at other Wyckoff positions. We note that (E.5) does not give the full symmetry indicator classification in space group 225 [47,48], and the $\mathbb{Z}$ invariant given by the mirror Chern number also remains well defined and contains additional information.

## E.2  Rotation eigenvalues

Another way to formulate the bulk invariant relies on the Chern-number being expressible through the difference in the occupied rotation eigenvalues at the rotation-invariant points $\boldsymbol{k} = \boldsymbol{0}$ and $\boldsymbol{k} = \infty$ [21,49]:

$$C = \sum_{n \in \mathrm{occ}} \left( \langle n(\infty) | S_z | n(\infty) \rangle - \langle n(\boldsymbol{0}) | S_z | n(\boldsymbol{0}) \rangle \right), \tag{E.6}$$

where $S_z$ is the generator of rotations around the $z$ axis and the Chern-number is calculated in the $k_z = 0$ plane (other orientations give equivalent results). To formulate the mirror Chern number, we insert $-iM_z$, which adds a $\pm 1$ prefactor to the mirror-even/odd states:

$$C_M = -\frac{1}{2} \sum_{n \in \mathrm{occ}} \left( \langle n(\infty) | iM_z S_z | n(\infty) \rangle - \langle n(\boldsymbol{0}) | iM_z S_z | n(\boldsymbol{0}) \rangle \right). \tag{E.7}$$

In general $M_z = \mathcal{I} \exp(i\pi S_z)$, in the spin-1/2 case this simplifies to $M_z = i\mathcal{I}\sigma_z$, hence $-iM_z S_z = \frac{1}{2}\mathcal{I}$. Substituting this, we find

$$C_M = \frac{1}{4} \sum_{n \in \mathrm{occ}} \left( \langle n(\infty) | \mathcal{I} | n(\infty) \rangle - \langle n(\boldsymbol{0}) | \mathcal{I} | n(\boldsymbol{0}) \rangle \right) = -\nu_I. \tag{E.8}$$

For general spin, using that $\mathcal{I}$ commutes with the spin operators, after some algebra we find

$$\begin{aligned}
C_M &= \frac{1}{4} \sum_s (-1)^{s-\frac{1}{2}} \sum_{n_s \in \mathrm{occ}_s} \left( \langle n_s(\infty) | \mathcal{I} | n_s(\infty) \rangle - \langle n_s(\boldsymbol{0}) | \mathcal{I} | n_s(\boldsymbol{0}) \rangle \right) \\
&= \sum_s (-1)^{s+\frac{1}{2}} \left( s + \frac{1}{2} \right) \nu_I^s.
\end{aligned} \tag{E.9}$$

As we saw, in the spin-1/2 case studied in detail, Eqs. (17, (E.7), and (E.1)) are all equivalent formulations of the same invariant, as demonstrated by their equivalence for different values of the chemical potential [Fig. E.1(a)].

## F  Amorphous network model

In order to ensure four-fold coordination of each node of the amorphous network, we generate the network following the method described in Refs. [21, 30], which creates a

graph by generating $N$ random lines on a plane, with $N$ chosen from a Poisson distribution whose mean is set to $2R\sqrt{\pi\rho}$, with $\rho$ the chosen density of the graph and $R$ the outer radius of the network. The angle and offset of the lines is uniformly distributed in $[0, 2\pi)$ and $[0, R]$ respectively. We define the intersections of each pair of lines as a network node. We ensure the two-in-two-out pattern of propagating modes at each node by orienting the links in an alternating fashion along each of the straight lines. There is no dependence of the scattering matrices on the length of the network links.

The graph is cut into an annulus shape by removing all of the nodes beyond the outer radius $R$ and within the inner radius $r$. This ensures periodic boundary conditions along the polar angle coordinate. In order to maintain four-fold connectivity in the bulk of the graph, the nodes outside of the network that are connected to nodes inside of the network are changed into sinks or sources, that either absorb modes from the network or emit modes to the network. The conductivity of the amorphous network is calculated by $g = G \ln(R/r)/2\pi$, with $G = (e^2/h) \sum_{i,j} |S_{ij}|^2$, $S_{ij}$ being the matrix element of the scattering matrix that connects the incoming modes originating from external sources beyond the network's outer edge to the outgoing modes exiting the network from its inner edge. A relaxation of the graph for visual clarity is optionally performed by averaging each node position to the center of its neighbors' positions.