# Peer review of "Isotropic 3D topological phases with broken time reversal symmetry"

_SciPost Physics_

## Round 1 · Referee Report · Anonymous (Referee 1) · 2025-8-1

Strengths

1- Presents a model of an amorphous system with broken time-reversal symmetry that retains isotropy on average and can host topologically protected phases.

2- The numerical results reveal interesting physical behavior.

Weaknesses

1-Crucial steps in the construction are delegated to software packages and prior work by the authors, leading to gaps and abrupt transitions in the presentation.

2-Creates the impression that the work is an incremental advance, relying heavily on previously developed insights.

3-Presentation unclear about the status and relation of the many different model variants.

4-Discussion of results is very brief, and again lacks clarity about the relevant model aspects.

Report

This work presents a model of an amorphous system with broken time-reversal symmetry that retains isotropy on average and can host topologically protected phases. The central idea of the paper is timely, and the numerical results reveal some interesting physical behavior. However, crucial steps in the construction are delegated to software packages and prior work by the authors, leading to gaps and abrupt transitions in the presentation. Thereby, the work also creates the impression that it is an incremental advance, relying heavily on previously developed methods and insights. In general, the paper makes it very difficult to identify key points in the construction of the models and their relation to the general and specific results. This is detailed below in the list of requested changes.

Requested changes

Main clarifications about the model:

1) The authors should clarify which features of the model are truly minimal. Is it sufficient to have distance-dependent complex hoppings as in Eq. (5), possibly with a simple constraint? In particular, can a more general criterion for the scalar TRS breaking nature of these hoppings be given, beyond its origin from the scalar quantity (∇×M) · r in the microscopic model? The authors state: "The assumption of spatial isotropy constrains the dependence of Hhopping on the hopping vector, satisfied for example by H_hopping s−p found in the previous section.", so they seem to have a broader criterion in mind.

If on the other hand any complex hopping where the phases cannot be gauged away suffices, this should be clearly stated, ideally early on, before one has to understand the intricacies of the microscopic model.

2) Related: The microscopic model is completely discarded when it comes to the actual results in Section 2.2. How important is the specific realization of these hoppings via chiral magnetic clusters?

3) Reproducibility of the perturbative calculation: The paper merely states: "We use second-order quasi-degenerate perturbation theory (assisted by the Python software package Pymablock [12]) to obtain the effective hopping tsp between the s and p1/2 orbitals." What steps must a reader follow to reproduce these hoppings?

4) The statement "different distance dependence of the microscopic hopping amplitudes from the px and py,z orbitals." should be clarified in the diagrams. It should also be distinguished from the distance dependence due to hopping to next-nearest or to third-nearest-neighbors. (App D states "third-nearest-neighbor s–p hopping [Fig. 1(b)]." but this Figure does not indicate these hoppings.)

5) The paper states: "and f is a complex function of the hopping distance." Do the authors mean a specific function (if so, which one?), or would any function do for this purpose?

6) "Rather than simulating an amorphous system with two families of atoms and two degrees of freedom per atom, for simplicity and without loss of generality we simulate one type of atom with four degrees of freedom." I understand the points of simplicity and that this does not lose generality. However, this further deemphasize the value of the earlier considerations of the microscopic model. At many points onwards, it is difficult to find out which model is then discussed, especially since detailed figure descriptions are in the Appendix, and this contains yet other model variants, such as in Appendix D. For instance: At the start of 2.2 the authors introduce yet another, effective model, Heff. Is this now obtained from the 4x4 models of Appendix A, the amorphous realization (5) of the microscopic model, ...?

Main clarifications about the results: 7) All figures: Beyond stating plot details are in App B, it would be useful to state at least from which model it is obtained, and what can be learned from these results.

8) Bulk invariants. The abstract states that a bulk invariant will be "constructed". Is the invariant used in 2.2.1 a new one, or in [10,13] or elsewhere (is the reference at the end for the definition of a compactified mirror-invariant plane, or for the invariant itself? In the latter case more standard citation convention would be to put the reference just before the equation, and the word "constructed" would not apply).

9) In the discussion the authors state: "We found results consistent with critical scaling, deviations from which are likely due to finite-size effects." but these points are not discussed in detail in the results section.

Due to these issues, a more detailed assessment of the specific results is difficult, as the paper is too vague, beyond which model has been applied, on which of the ingredients then are then relevant for the physics. This is not helped by the brevity of the discussion.

Minor points. 10) In the abstract the authors call the model "fully" isotropic, even though it is only isotropic on average/in the ensemble. This distinction is important and is, in fact, one of the main interesting aspects of these systems.

11) The beginning of subsection 2.1.2 states "Based on the symmetry-allowed terms of the continuum model (1), we now construct a microscopic model that preserves isotropy while breaking TRS.", but this subsection only proceeds up to the rock salt implementation, while the statistically isotropic model is only constructed in 2.1.3. The main purpose of 2.1.2 seems to be the construction of scalar TRS breaking hoppings.

12) The references appear inconsistent. General context example: "This topological phase is analogous to crystalline mirror- Chern insulators" - it would be pertinent then to cite papers describing these (see also point 8 above). Why is [16] cited for the kernel polynomial method? If it contains anything of specific relevance for this paper, it should be mentioned. In contrast, only in Appendix C we learn that some key ideas about the amorphous models used here have already been developed in the earlier work [34] involving one of the authors. To fully describe its significance, the paper would reach further, e.g., seek context with works such as PRX 13, 031016 (2023).

If some of the main clarifications are already addressed elsewhere in the paper, the authors should consider restructuring the manuscript to make this more apparent. These are not just questions of presentation-they directly impact on the reproducibility of the results.

Recommendation

Ask for major revision

---

## Round 1 · Referee Report · Anonymous (Referee 2) · 2025-8-27

Strengths

1) This work provides an explicit example of a 3D amorphous phase different from previously predicted ones. 2) It approaches the problem in a thorough way combining symmetry analysis, topological invariants and transport calculations in amorphous systems. 3) It clearly demonstrates the existence of an isotropic topological phase with mirror Chern number C=1.

Weaknesses

1) The "scalar magnetism" has been considered before and the authors do not discuss the previous literature. 2) There is a paradox in the results when comparing transport and spectral functions for the C=2 case which is not explained sufficiently. 3) Some mathematical derivations are not clear.

Report

In this work, the authors present a proposal for a magnetic, amorphous topological insulator where the magnetic order breaks only time-reversal symmetry but no spatial symmetry (a type of scalar magnetism). To do so, the authors first write down a crystalline model with scalar magnetism which shows surface Dirac points protected by mirror symmetry only in mirror preserving surfaces (a 3D mirror Chern insulator). Using the same local hopping implementation of scalar magnetism, the authors then write down an amorphous version of such state which preserves average mirror symmetry in every direction, and show that the corresponding surface Dirac cones appear in the spectral function for any surface. Finally, the authors double the model to show the protection of two Dirac cones in the spectral function but surprisingly not in transport (there is localization in this case).

In my opinion the work is interesting because it provides an explicit example of a 3D amorphous phase different from previously predicted ones. However, there is room for improvement in the organization and presentation of the results. In general, I encourage the authors to write more explicit equations instead of words when possible, especially when trying to describe mathematical features of the proposal. I also present some concerns that I believe should be answered before this manuscript is suitable for publication.

  • First, a general comment: The author’s proposal of a scalar magnetic order parameter is known in the literature as the magnetic toroidal monopole (see for example the review by Hayami, Symmetry 16, 926 (2024), or Hayashida, Advanced Materials 37, 2414876 (2025)). Toroidal multipoles have an extra minus sign under inversion compared to regular multipoles. The normal electric monopole is the charge (P-even, T-even), the magnetic monopole is P-odd, T-odd, the toroidal electric monopole is P-odd, T-even (it represents scalar chirality) while the toroidal magnetic monopole is P-even, T-odd, and it is the magnetic scalar the authors are after. In the discussion the authors mention the magnetoelectric response P. E cross B, which is exactly the response expected for a material with magnetic toroidal monopole order. I believe the manuscript would connect with a broader readership by reviewing this nomenclature since there are materials with this type of order (which the authors could also review). In relation to this, the authors could mention the space groups of the crystalline tight binding models and the magnetic space group of the model with complex hoppings.

  • The most important physics question to me is this: The authors show that the amorphous model with C_M = 2 does show two Dirac points, which suggests a Z classification. However, transport calculations in this case show localization, which suggests that the classification is Z2. The authors then suggest that a local, symmetry preserving perturbation should be able to open a spectral gap too. Does this mean the authors did search for such a perturbation and did not find it? If such a perturbation were to exist, wouldn’t this be resolved by computing the invariant (6) in the doubled model? The sentence “We constructed a bulk Z invariant […] indicating the presence of a protected ungappable surface Dirac cone for odd values” is very confusing. If the mirror Chern number is Z, it protects any number of Dirac cones. Otherwise the invariant is not Z. Or there is something in the classification of amorphous systems which does not work in the same way as crystalline systems, but then it should not be called a “bulk Z invariant”. In my opinion, the authors should use any of their proposed methods to compute invariants to do so for the doubled model, and comment on these paradox.

  • About the order of presentation: The microscopic origin of the phase dependent hopping from virtual hopping through a cluster of magnetic atoms does not appear very important for the rest of the text, and makes the symmetry analysis and the arguments about spin splitting harder to follow. The four band crystalline model is described in the section “microscopic implementation” but it is not microscopic but effective. I think a self contained description of four band models (continuum, lattice, amorphous) with their symmetries would be most easily understood, with a separate section, possibly an appendix, for the microscopic proposal to generate the phase dependent hopping.

  • In Section 2.1.1, it is stated that the continuum model has time-reversal-like symmetry if restricted to order k^2. Can the authors explicitly write down what this symmetry is with an equation S H S = H? And how do the cubic terms break it down? I am unable to follow the discussion of this effective time-reversal through the text, and I believe many readers will too.

  • Later on: “The hopping phase is distance dependent due to the different distance dependence of the microscopic hopping amplitudes from the px and py,z orbitals. This ensures that the hopping phase cannot be removed by a global basis-transformation introducing a relative phase between the s and p wavefunctions”. Can the authors write what distance dependence they mean? And how does this ensure the breaking of T?

  • Similarly, in Appendix D, the authors write “The TRS-breaking terms of our model are next-next-nearest neighbor terms, which leads to linear TRS-breaking terms intrinsically cancelling out and only cubic terms remaining”. This was also unclear to me. Here linear and cubic means expanding in k? And for some reason the t3 terms do not lead to k^3 terms, but the t4 terms do? Again equations would help.

  • At the end of section 2.1.2, the authors claim to “provide a minimal spin model” that realizes the magnetic texture needed to make the desired complex hopping. But the authors do not write any equation at all. I think if the authors deem this important, they should write down the model explicitly and explain their claims with equations. Otherwise, they may consider removing this paragraph.

Requested changes

See report

Recommendation

Ask for major revision

---

## Editorial Decision

awaiting_resubmission